# Learning Fast and Slow: Representations for In-Context Weight Modulation

**Andrey Zhmoginov** [1]  **Jihwan Lee** [1]  **Max Vladymyrov** [1]  **Mark Sandler** [1]

## Abstract

Most natural sequential processes involve a spectrum of different time scales: from fast-changing variations responsible for local structure to slowly-changing dynamics akin to memory that captures context information. Here we propose a method for learning such disentangled slow-fast representation in activations of a conventional Transformer model. We accomplish this by employing regularization techniques inspired by contrastive learning. This proposed approach can be further analyzed by adopting a Gaussian process prior resulting in a Variational Autoencoder interpretation of a Transformer model. We evaluate our techniques on synthetic in-context learning tasks and widely-used text benchmarks, where we show the emergence of disentangled representations. We then propose a HyperNetwork-inspired approach, where the slow representations are employed to modulate the weights of the transformer performed on the fast short-range activations. We demonstrate that adding such modulation makes it possible to generate models specialized to a particular context on the fly.

## 1. Introduction

Sequential data across a variety of domains, including physical measurements, audio, language, video and numerous other types, often exhibit rich temporal spectra (Dieterich, 2002; Hu et al., 2016; Birnbaum et al., 2019). The generative processes underlying different components of these signal frequently produce distinct spectral components, each reflecting different mechanisms and carrying different types of information. For example, a speech recording of a dialogue contains higher-frequency audio signals capturing audio details, while lower-frequency latent features encode phonemes, then utterances. The lowest-frequency components may carry information about speaker's identity, or

gradual changes in conversation topic. Similarly, in a text domain, we expect to find slowly-evolving representations encoding information about the current topic, text sentiment and overall context. Another important example that we address in this paper is few-shot in-context learning. Here, the nature of the task is expected to be gradually revealed as more and more examples are being processed. Hence, we can think of the task representation as a variable that slowly changes along the sequence and eventually saturates when the task is fully specified.

In this publication, we investigate the unsupervised disentanglement of different time scales in sequences and time series. Specifically, we focus on enhancing the emergence of slow-evolving features within the activations of a standard Transformer model (Vaswani et al., 2017) by incorporating novel auxiliary regularization techniques. We demonstrate the effectiveness of this technique and show that resulting disentangled representations effectively capture both global and local information about the sequence. This allows us to interpret the computation performed by the model in a few-shot in-context learning setting. We illustrate the process in which slow features stabilize and saturate as the Transformer processes an increasing number of examples, and show that these slow features end up reflecting the information about the underlying task showcased with given examples.

Recalling that slow features frequently characterize the global context, we then propose to modify the Transformer model to explicitly decouple the influence of these slow degrees of freedom on the local computation. This allows us to view a trained sequence model as an entire "manifold" of sequence models all specialized to particular contexts (that is dynamically discovered by the model as it is processing the sequence). We then show that by freezing slow representations we can recover lightweight models uniquely suited for performing one specific task, or working in a specific context. This process is performed in a single step and does not require any tuning. Furthermore, the resulting model no longer needs a prompt or few-shot demonstrations in context.

This paper is structured as follows. In Section 2, we discuss related work listing publications exploring similar research topics. Then, in Section 3, we outline two core components of our method: (a) regularization techniques leading to

---

[1] Google Research, now Google DeepMind. Correspondence to: Andrey Zhmoginov <azhmogin@google.com>.

*Proceedings of the 1st Workshop on In-Context Learning at the 41st International Conference on Machine Learning*, Vienna, Austria. 2024. Copyright 2024 by the author(s).

emergence of slow features in Transformer activations and (b) additional architectural elements that allow these slow features to have an expressive modulation effect on the local computation performed by the model. Section 4 starts with a discussion of datasets that we use in our experimental setup and then covers our results obtained with these datasets. Finally, in Section 5, we outline our conclusions.

## 2. Related Work

**Learning representations with various scales.** Publications (Xu et al., 2022; Tang et al., 2022; Rao et al., 2021; Chen et al., 2023) have explored techniques for assessing the significance of individual tokens with varying levels of detail, aiming to reduce computational overhead. Specifically, (Xu et al., 2022) shares a conceptual similarity with our approach, which involves applying distinct update mechanisms to tokens based on their importance. While their approach distinguishes between informative and placeholder tokens, ours divides embedding dimensions into two segments, each tasked with capturing either local or global context.

**Transformer + VAE.** Integrating Transformer and Variational Autoencoder (VAE) (Kingma & Welling, 2014) has been a subject of numerous endeavors. (Casale et al., 2018) employs Gaussian processes as priors for the latent space, enabling the model to capture intricate data dependencies. Addressing the issue of controllability in narrative generation, (Wang & Wan, 2019; Fang et al., 2021) develop a conditional VAE framework. (Henderson & Fehr, 2023) introduces a model that incorporates nonparametric variational methods to enhance the information bottleneck in Transformers, leading to better capture of latent representations and improved efficiency across various natural language processing tasks. Similarly to the previous work, our approach proposes a VAE-based method with a meticulously designed regularizer, enabling more flexible control over the representations, ensuring they evolve slowly.

**In-context learning.** In-context learning has garnered significant attention among researchers, particularly with the rise of large language models, owing to its adaptability to unforeseen tasks (Brown et al., 2020). Several studies (Von Oswald et al., 2023; von Oswald et al., 2023; Liu et al., 2022; Min et al., 2022; Zoph et al., 2022) have examined the mechanics of in-context learning to grasp its functionality and rationale. Our proposed slow-fast representation learning approach is able to effectively bolster in-context learning for generative models by effectively capturing both global and local contexts across diverse tasks.

## 3. Method

In this section, we detail two core components of our method: (a) learning a slow-evolving global context representation in a sequence, and (b) using this representation to adjust model weights for local computation. The outline can be summarized as follows:

1. We discuss *slow-evolving representations* in Section 3.1 and introduce element-wise regularization techniques ($\mathcal{R}_C$ and $\mathcal{R}_D$) in Section 3.2;

2. We extend these techniques to Gaussian process priors in Section 3.3, proposing a *VAE-based approach* (can be used in place of the above regularization);

3. In Section 3.4, we explore using slow features to *modulate local computation* and present a novel architecture for context-specific model generation;

4. Finally, in Section 3.5, we cover an optional *auxiliary loss* $\mathcal{L}_{\mathrm{aux}}$ and additional *feature augmentations* to further disentangle fast-slow representations thus enhancing model performance.

### 3.1. Learning Slow Features

Consider a conventional Transformer model processing an input sequence $\boldsymbol{t} := (\boldsymbol{t}_1, \ldots, \boldsymbol{t}_n)$, where each $\boldsymbol{t}_i$ represents a discrete token. We follow a convention, where all "processing stages" (that we later enumerate with $\nu$) including self-attention and MLP layers have residual connections. Assuming that the unknown generative process that outputs $\boldsymbol{t}$ incorporates slowly changing latent variables, we may attempt to regularize our Transformer model with the goal of identifying such slow features as a part of the computation. Here we use the term "slowly" informally, but each regularizer described below in effect defines it implicitly. We present a more detailed discussion of this subject in Appendix A.

In some circumstances, our prior knowledge of the unknown generative process could be very detailed and include, for example, information about an entire spectrum of time scales involved. Here, however, we attempt to make just a few simplifying assumptions. For example, while it is possible to regularize activations at multiple stages of computation, in the following we target a single stage $\nu = \ell$. We then partition model activations $\boldsymbol{z}^\ell$ at this stage into just two parts $\boldsymbol{z}_i^\ell := (\boldsymbol{x}_i^\ell, \boldsymbol{y}_i^\ell)$ with activation components $\boldsymbol{x}^\ell$ assumed to be completely unconstrained and using a specially designed regularizer $\mathcal{R}$ for incentivizing $\boldsymbol{y}^\ell$ to change slowly.

### 3.2. Slow Features: Element-Wise Regularizers

The simplest regularizer that enforces continuity in $\boldsymbol{y}^\ell$ can simply penalize large time step differences in $\boldsymbol{y}_i^\ell$, or in its

normalized value, for instance we consider:

$$\mathcal{R}_C^\ell \sim \sum_{s=2}^{n} \left\| \boldsymbol{n}_s^\ell - \boldsymbol{n}_{s-1}^\ell \right\|^2,$$

where $\boldsymbol{n}_i^\ell := \boldsymbol{y}_i^\ell / \|\boldsymbol{y}_i^\ell\|$. In the rest of this section and in Sec. 3.3, we drop $\ell$ for brevity.

In practice, adding the regularization term into the loss with some non-zero weight $w_C$ can incentivize the model to generate constant activations $\boldsymbol{y}_i$ with $\mathcal{R}_C = 0$, at least locally. A common way of stopping $\boldsymbol{y}$ from collapsing to a constant value is to adopt some form of contrastive learning approach. For example, inspired by the orthogonal projection loss (Ranasinghe et al., 2021), we regularize the scalar product of activations across samples in the batch for each sequence element independently:

$$\mathcal{R}_D \sim \sum_{s, \alpha, \beta} \left( \boldsymbol{n}_s^{(\alpha)} \cdot \boldsymbol{n}_s^{(\beta)} - \delta_{\alpha, \beta} \right)^2,$$

where $\alpha$ and $\beta$ are indices of two samples in the batch and $\delta_{\alpha, \beta}$ is the delta function. We refer to regularizers that do not depend on cross-element correlations as *element-wise*. This particular regularizer is designed to favor orthogonality of sample representations within the batch and it proved to be sufficiently effective in our experiments, where we end up optimizing the joint loss $\mathcal{L}' = \mathcal{L} + w_C \mathcal{R}_C + w_D \mathcal{R}_D$.

There is also an alternative approach to incentivizing representation diversity that is based on parameter estimation that we discuss in detail in Appendix B. Detailed comparison with this approach will be the subject of our future work.

### 3.3. Slow Features: VAE-Based Approach

A more principled approach that extends the element-wise regularization method and gives us a more nuanced control over the characteristics and smoothness of $\boldsymbol{y}$ is to view the Transformer as a Variational Autoencoder model (Kingma & Welling, 2014) and choose a Gaussian process prior for $\boldsymbol{y}$. More specifically, we assume that a prior over $\boldsymbol{y}$ is a multivariate Gaussian distribution $p_\circ(\boldsymbol{y}_1, \ldots, \boldsymbol{y}_n)$ with $\langle \boldsymbol{y}_s \rangle = \boldsymbol{\mu}^y$ and covariance $\langle (\boldsymbol{y}_s - \boldsymbol{\mu}^y)(\boldsymbol{y}_t - \boldsymbol{\mu}^y) \rangle$ being given by a known kernel $\mathcal{K}_{s,t} = \mathcal{K}(|s - t|)$. This prior ensures that $\boldsymbol{y}$ do not degenerate becoming constant and that computed at two nearby points in time, these activations maintain a certain degree of coherence defined by $\mathcal{K}$.

In this setup, we assume that the probability distribution over $\boldsymbol{t}$ can be represented as $\int p_\phi(\boldsymbol{t}|\boldsymbol{x}, \boldsymbol{y}) p_\circ(\boldsymbol{x}, \boldsymbol{y}) \, d\boldsymbol{x} \, d\boldsymbol{y}$ with $p_\phi(\boldsymbol{t}|\boldsymbol{z})$ being a causal *decoder* and $p_\circ(\boldsymbol{x}, \boldsymbol{y}) = p_\circ(\boldsymbol{x}) p_\circ(\boldsymbol{y})$ being the prior. Following a conventional Variational Autoencoder setup, we can then use a variational approximation $q_\psi(\boldsymbol{t}|\boldsymbol{z})$ of $p_\phi(\boldsymbol{t}|\boldsymbol{z})$ and employ the evidence lower bound (ELBO) to derive a VAE objective:

$$\mathcal{L} = \mathbb{E}_{\boldsymbol{t} \sim p(\boldsymbol{t})} \Big[ \mathbb{E}_{\boldsymbol{z} \sim p_\phi(\boldsymbol{z}|\boldsymbol{t})} \log q_\psi(\boldsymbol{t}|\boldsymbol{z}) + \\ + D_{\mathrm{KL}}(p_\phi(\boldsymbol{z}|\boldsymbol{t})|p_\circ(\boldsymbol{z})) \Big].$$

This formulation allows us to view the full Transformer model as a combination of two parts: an encoder $q_\psi(\boldsymbol{z}|\boldsymbol{t})$ mapping the input $\boldsymbol{t}$ to intermediate activations $\boldsymbol{z}$ at some layer $\ell$, and a *decoder* $p_\phi(\boldsymbol{t}|\boldsymbol{z})$ reconstructing the input from these latent variables. Choosing causal Transformer layers for parameterizing $p_\phi$ and $q_\psi$, we see that the only difference of our model from a conventional Transformer is the fact that the activations $\boldsymbol{z}^\ell$ are no longer deterministic.

For simplicity, we assume statistical independence of $\boldsymbol{x}$ and $\boldsymbol{y}$ in both $p_\phi(\boldsymbol{z}|\boldsymbol{t})$ and $p_\circ$ and adopt the $\beta$-VAE approach relying on two independent constraints, on $\boldsymbol{x}$ and $\boldsymbol{y}$ resulting in:

$$\mathcal{L} = \mathbb{E}_{\boldsymbol{t} \sim p(\boldsymbol{t})} \Big[ \mathbb{E}_{\boldsymbol{z} \sim p_\phi(\boldsymbol{z}|\boldsymbol{t})} \log q_\psi(\boldsymbol{t}|\boldsymbol{z}) + \\ + \beta_x D_{\mathrm{KL}}(p_\phi(\boldsymbol{x}|\boldsymbol{t})|p_\circ(\boldsymbol{x})) + \beta_y D_{\mathrm{KL}}(p_\phi(\boldsymbol{y}|\boldsymbol{t})|p_\circ(\boldsymbol{y})) \Big]. \tag{1}$$

While it could be useful to define a prior on $\boldsymbol{x}$, in the following we choose $\beta_x = 0$ and let $\boldsymbol{x}$ being unconstrained, only constraining our slow activations $\boldsymbol{y}$. As a result, we can view the first term in Eq. (1) as a conventional autoregressive sequence reconstruction loss, while the last KL divergence term acts as a regularizer on $\boldsymbol{y}$ and is described below.

Here we derive $D_{\mathrm{KL}}$ loss for a scalar sequence $y_i$, later generalizing this derivation to vector-valued sequences. The Gaussian process prior on $y$ is defined by specifying $\mu^y$ and the kernel $\mathcal{K}$. In our model, we chose $\mu^y = 0$ and $\mathcal{K}_{i,j} = K^2(\nu \delta_{i,j} + (1 - \nu) \mathcal{K}_{i,j}^{\mathrm{RBF}})$ with some constant $K > 0, \nu \in [0, 1]$ and a Radial Basis Function (RBF) kernel $\mathcal{K}_{i,j}^{\mathrm{RBF}} = \exp(-\|i - j\|^2 / 2\sigma^2)$, where $i$ and $j$ are two sequence positions. As a result, our kernel is parameterized by just two scalars: $\nu$ controlling a mixture of delta-correlated noise and the RBF kernel and $\sigma$ that controls the smoothness of $y$. To simplify our calculations further, we define our encoder $p_\phi(y|\boldsymbol{t})$ as

$$p_\phi(y|\boldsymbol{t}) \propto \exp \left[ -\sum_i \frac{(y_i - \mu_i(\boldsymbol{t}))^2}{2\sigma_i(\boldsymbol{t})^2} \right],$$

effectively treating elements $y_i$ taken at different positions as statistically independent draws from corresponding Gaussian distributions. This, of course, guarantees that $D_{\mathrm{KL}}$ can never reach zero, but it is still expected to adequately regularize our encoder $\boldsymbol{t} \to \boldsymbol{y}$. We can then easily compute the KL divergence of this distribution with our Gaussian

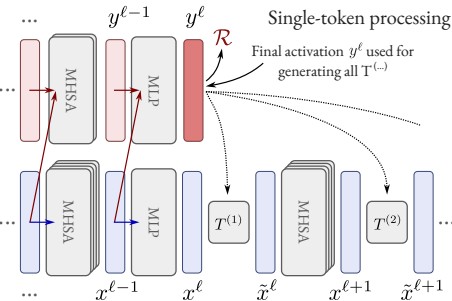

*Figure 1.* Model architecture showing processing of a single token with the activation component $\boldsymbol{y}^{\nu+1}$ being a function of $(\boldsymbol{x}^\nu, \boldsymbol{y}^\nu)$ and $\boldsymbol{x}^{\nu+1}$ being a function of $\boldsymbol{x}^\nu$ alone; activations $\boldsymbol{y}^\ell$ are regularized to be slow-changing (via $\mathcal{R}$) and are then parameterizing transformations $\mathbf{T}^{(q)}(\cdot; \boldsymbol{y}^\ell)$ mapping $\boldsymbol{x}^\nu$ to $\tilde{\boldsymbol{x}}^\nu$ for $\nu \geq \ell$.

process prior:

$$2D_{\mathrm{KL}} = \log|\mathcal{K}| - \sum_{i=1}^{n} \log \sigma_i - n + $$
$$+ \sum_{i,j=1}^{n} \mathcal{K}_{i,j}^{-1} \mu_i \mu_j + \sum_{i=1}^{n} \mathcal{K}_{i,i}^{-1} \sigma_i.$$

Notice that $\log|\mathcal{K}| - n$ is a constant and $\mathcal{K}^{-1}$ can be pre-computed making this calculation sufficiently low-cost. For vector-valued $\boldsymbol{y}$, the kernel becomes block-diagonal and the regularizer can be seen to have the same general form with an additional outer summation over different components of $\boldsymbol{y}$.

The complete model can then be seen as a combination of a conventional autoregressive reconstruction loss and a regularization term $D_{\mathrm{KL}}$. The latter can be seen to penalize very large and very small values of $\sigma_i$ and non-zero $\mu_i$. The regularization effect on $\mu$ can be studied by computing eigenvectors of $\mathcal{K}^{-1}$. For a sufficiently large $\sigma$, the eigenvalues can typically be seen to grow rapidly with the number of oscillations in the corresponding eigenvectors, highlighting the fact that this regularization term suppresses rapidly changing fluctuations. Also notice that the encoder mapping $\boldsymbol{t}$ to $\boldsymbol{z}$ is now probabilistic with a deterministic unconstrained $\boldsymbol{x}(\boldsymbol{t})$ and $\boldsymbol{y}(\boldsymbol{t}) \sim \mathcal{N}(\boldsymbol{\mu}_i, \mathrm{diag}\,(\boldsymbol{\sigma}_i))$. In practice, vectors $\boldsymbol{\mu}_i$ and $\boldsymbol{\sigma}_i$ can be generated in our Transformer via a composition of self-attention and MLP layers as conventional activations. In effect, the overall setup can be seen as a balancing game between the reconstruction objective and injecting just enough noise in $\boldsymbol{y}$ while penalizing $\boldsymbol{\mu}_i$ for being non-zero and fast changing.

### 3.4. Computation Modulation by Slow Features

Assuming that our regularization enforces changes in $\boldsymbol{y}^\ell$ sufficiently slowly, we observe that these activations influence subsequent operations (typically matrix multiplications

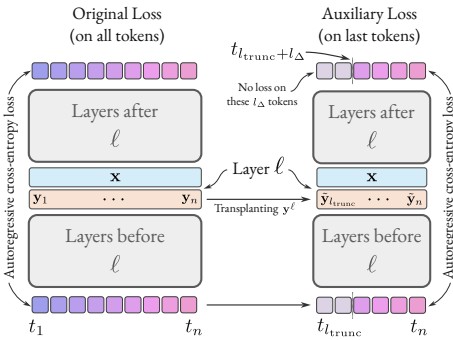

*Figure 2.* Auxiliary loss incentivizing the Transformer to encode long-range information in $\boldsymbol{y}$: the auxiliary loss is applied to an input sequence truncated at $l_{\mathrm{trunc}}$ with the autoregressive loss applied to all, but the first $l_\Delta$ tokens, the slow variables $\boldsymbol{y}^\ell$ are borrowed from the Transformer running on the original sequence (with a possible smoothing augmentation mapping $\boldsymbol{y}$ to $\tilde{\boldsymbol{y}}$).

within the ensuing self-attention or MLP layer) essentially by adding a slowly varying bias term. This happens because $\mathbf{W}^\ell \boldsymbol{z}_i^\ell$ can be expressed as $\mathbf{W}_x^\ell \boldsymbol{x}_i^\ell + \tilde{\boldsymbol{b}}_i$, where $\mathbf{W}$ is a linear operator and $\tilde{\boldsymbol{b}}_i := \mathbf{W}_y^\ell \boldsymbol{y}_i^\ell$ is evolving slowly with $i$. In other words, we can think of components $\boldsymbol{y}^\ell$ as discovered slow features that modulate the computations done on the bulk of the model activations $\boldsymbol{x}^\ell$.

Since changing layer biases may not be sufficiently expressive, we introduce a more complex mechanisms that allows $\boldsymbol{y}^\ell$ to affect both biases and weights of the computation over $\boldsymbol{x}$ at multiple following layers. Furthermore, for the purposes of discovering specialized models (as will be discussed below), we decouple the computation on $\boldsymbol{x}^\nu$ from $\boldsymbol{y}^\nu$ except for explicit $\boldsymbol{y}^\ell$-dependent transformations applied to $\boldsymbol{x}^\nu$ for $\nu \geq \ell$ (see Fig. 3.3). To be more specific, in all computation stages $\nu \leq \ell$, we make sure that $\boldsymbol{x}^\nu$ are only dependent on $\boldsymbol{x}^{\nu-1}$, while $\boldsymbol{y}^\nu$ are allowed to depend both on $\boldsymbol{y}^{\nu-1}$ and $\boldsymbol{x}^{\nu-1}$. This also requires that we process $\boldsymbol{x}^\nu$ and $\boldsymbol{y}^\nu$ using separate self-attention heads. For $\nu \geq \ell$, before processing each activation $\boldsymbol{x}^\nu$, we apply an additional transformation to it producing $\tilde{\boldsymbol{x}}^\nu = \mathbf{T}^\nu(\boldsymbol{x}^\nu; \boldsymbol{y}^\ell)$, which we then use in the following MLP or self-attention layers. The transformation $\mathbf{T}^\nu(\boldsymbol{x}; \boldsymbol{y}^\ell)$ is:

$$\mathbf{T}^\nu(\boldsymbol{x}; \boldsymbol{y}^\ell) := \boldsymbol{x} + \delta\hat{\mathrm{W}}^\nu(\boldsymbol{y}^\ell)\boldsymbol{x},$$
$$\delta\hat{\mathrm{W}}_{ij}^\nu(\boldsymbol{y}^\ell)L = \sum_k \hat{\mathrm{L}}_{ik}^\nu(\boldsymbol{y}^\ell)\hat{\mathrm{R}}_{jk}^\nu(\boldsymbol{y}^\ell),$$

where $\delta\hat{\mathrm{W}}^\nu(\boldsymbol{y}^\ell)$ is generated from:

$$\hat{\mathrm{L}}^\nu(\boldsymbol{y}^\ell) = \sum_{m=1}^{M} \hat{\mathrm{L}}^{\nu,(m)} \sigma_m^\nu(\boldsymbol{y}^\ell),$$
$$\hat{\mathrm{R}}^\nu(\boldsymbol{y}^\ell) = \sum_{m=1}^{M} \hat{\mathrm{R}}^{\nu,(m)} \sigma_m^\nu(\boldsymbol{y}^\ell),$$

where $\hat{L}^{\nu,(m)}$, $\hat{R}^{\nu,(m)}$ are additional learned matrices and $M$ is their total number. The nonlinearity $\sigma(\boldsymbol{y}^\ell)$ is typically chosen to be a hyperbolic tangent $\sigma^\nu(\boldsymbol{y}^\ell) = \tanh\left(\mathbf{S}^\nu \boldsymbol{y}^\ell\right)$, or a softmax $\sigma^\nu(\boldsymbol{y}^\ell) = \text{softmax}\left(\mathbf{S}^\nu \boldsymbol{y}^\ell\right)$ with $\mathbf{S}^\nu$ being a learned linear transformation.

### 3.5. Slow Feature Transplantation and Additional Techniques

Our approach for using $\boldsymbol{y}$ to modulate the computation on $\boldsymbol{x}$ may be viewed as a way of generating context-specialized models. Indeed, if $\boldsymbol{y}$ encodes information about the current context, we can freeze it (essentially keeping it constant along the sequence after some point) and "fold" generated near-identity transformations $\mathbf{T}(\boldsymbol{x}; \boldsymbol{y}^\ell)$ into the following matrix multiplications to produce a lightweight *context-specialized* model operating on $\boldsymbol{x}$ *alone*. Therefore, in effect, training a single modified Transformer model, we can learn an entire manifold of specialized compact models that can be isolated and frozen to only act in a given context.

For this technique to work, it is crucial that $\boldsymbol{y}$ encode only the global, slowly-changing context and not local information. Concurrently, it is also important that $\boldsymbol{x}$ does not absorb long-context dependencies. However, training Transformer using techniques described above may not be sufficient to guarantee such perfect disentanglement between $\boldsymbol{x}$ and $\boldsymbol{y}$. Thus, in our experiments, we used two additional techniques to further disentangle these features.

**Auxiliary loss.** The first idea is that $\boldsymbol{x}$ may be forced to rely on extracting global information from $\boldsymbol{y}$ alone if we can limit the model attention to a smaller local window. We accomplished this by adding a new *auxiliary loss* component $\mathcal{L}_{\text{aux}}(\bar{\boldsymbol{t}}; \bar{\boldsymbol{y}}^\ell)$ computed by a Transformer applied to a truncated sequence $\bar{\boldsymbol{t}}$ (see Fig. 3.3) with $\bar{\cdot}$ referring to tensors in this model. In most of our experiments, the slow variable sequence $\bar{\boldsymbol{y}}^\ell(\boldsymbol{t})$ was frozen and set to be equal to $\boldsymbol{y}^\ell$ pre-computed on the full sequence and truncated to match the time shift in $\bar{\boldsymbol{t}}$. The sequence $\bar{\boldsymbol{t}}$ was a truncated version of $\boldsymbol{t}_i$ with only $i \geq l_{\text{trunc}}$ elements kept for randomly sampled $l_{\text{trunc}}$. The autoregressive reconstruction loss $\mathcal{L}_{\text{aux}}$ was also only computed after $l_\Delta$ tokens thus giving local activations $\bar{\boldsymbol{x}}$ access to a local context of size $l_\Delta$, but communicating long-distance information via pre-computed $\bar{\boldsymbol{y}}^\ell$ alone. We observed that this technique was very efficient at forcing the model to rely almost solely on $\boldsymbol{y}$ for maintaining long-context information. Interestingly, this technique was also crucial for a proper operation of our VAE model since without this loss, VAE can typically ignore heavily regularized random $\boldsymbol{y}^\ell$ in favor of keeping all information in deterministically generated $\boldsymbol{x}^\ell$.

**$\boldsymbol{y}$ sequence augmentations.** We also used another technique for making it difficult for our model to keep local context information in $\boldsymbol{y}$. First of all, the injection of noise

in the VAE model naturally limits what it may communicate via $\boldsymbol{y}^\ell$, but a model with the element-wise regularizer can (and frequently does) "hide" local information in "slow" activations. Therefore, in our experiments with element-wise regularizers, we utilized random augmentations that smoothed and shifted the sequence $\boldsymbol{y}^\ell$ making it more difficult for the model to store local information in these slow features. The augmentations were either introduced in the forward path of the core model, or only used when generating $\bar{\boldsymbol{y}}^\ell$ for the auxiliary loss. Another augmentation we used extensively is to extend a value of $\boldsymbol{y}_i^\ell$ at position $i_* = l_{\text{trunc}}$ to the rest of the sequence ($i > i_*$). In other words, $\bar{\boldsymbol{y}}^\ell$ was held constant value throughout the entire sequence in the auxiliary loss. This augmentation can alone force $\boldsymbol{y}^\ell$ to be a slowly changing variable. More details about our full model can be found in Appendix C.

## 4. Experiments

### 4.1. Datasets

In this section, we describe two dataset families used in our experiments. The first dataset is *synthetic* with each sequence containing multiple arithmetic in-context learning tasks, each of which could be resolved approximately by solving a system of two linear equations. The second family which we call *text mixture* is based on frequently used `wikipedia` (Raffel et al., 2020) and `c4` (Foundation) datasets, where we combine two random excerpts to form a single training example.

**Synthetic In-Context Learning Setup.** Here we describe a simple synthetic in-context learning setup, where each individual sequence contains multiple in-context learning tasks. All sequences in our synthetic dataset contain $n_{\text{tasks}} \geq 1$ individual in-context learning *tasks*, each defined by its own two real-valued hidden parameters $(a_i, b_i)_{i=1}^{n_{\text{tasks}}}$. An individual in-context learning task $i$ is encoded via a sequence of $n_{\text{ex}}$ *examples* $\{\nu_{i,j} := h(\xi_{i,j}; a_i, b_i)\}_{j=1}^{n_{\text{ex}}}$, where $\xi_{i,j}$ are random example-specific arguments and $h$ is a function mapping these random arguments and task parameters into the actual example representation $\nu_{i,j}$.

In our dataset, $\xi_{i,j}$ are two $d$-digit integer numbers $A_{i,j}$ and $B_{i,j}$ with $d = 3$. Coefficients $a_i$ and $b_i$ are floating point numbers sampled uniformly from $[0, 10)$ and $[-9, 10)$ correspondingly. The actual sample $h(A_{i,j}, B_{i,j}; a_i, b_i)$ includes two $d$-digit arguments and a signed and truncated[1] $(d + 2)$-digit result $a_i A_{i,j} + b_i B_{i,j}$. In other words, we encoded linear combinations of two arguments with some unknown $a_i$ and $b_i$ and provided with several such examples, expected the model to perform this computation on entirely new $d$-digits numbers. In most of our experiments, we used $n_{\text{tasks}} = 4$ and provided $n_{\text{ex}} = 4$ examples for

---

[1] Not rounded, but using $\lfloor \cdot \rfloor$ instead

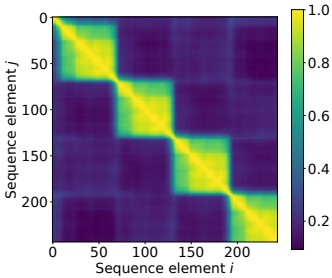

*Figure 3.* The dot product $\boldsymbol{n}_i \cdot \boldsymbol{n}_j$ plot for normalized $\boldsymbol{y}$ embeddings at two different locations in the sequence with 4 tasks and 4 examples per task.

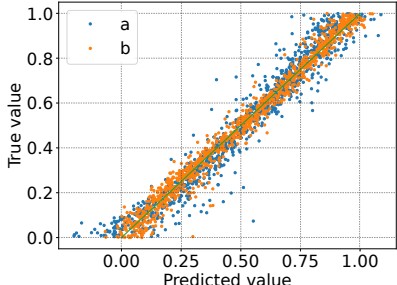

*Figure 4.* Linear regression results for the multipliers $a$ and $b$ given the average value of $\boldsymbol{y}$, the plot shows agreement between predicted and groundtruth values.

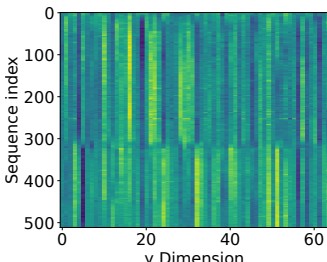

*Figure 5.* Evolution of $\boldsymbol{y}$ along the sequence containing two c4 text excerpts joined at 305.

each task. All examples were separated by a special token and all tasks within a sequence were separated by a different special token. A typical example with $a = 1$ and $b = 1$ could look like `012*023=+00035` and the same arguments for an example with $a = 0.5$, $b = -1.5$ would result in `012*023=-00028`. Examples of actual sequences are presented in Appendix D.

Since each task is specified by two unknowns $a$ and $b$, one typically needs 2 examples to find these multipliers. On top of that, since we only observe rounded results, there is an additional uncertainty in the reconstruction of $a$ and $b$. Therefore, we would expect a powerful model to reach a next-token-prediction accuracy close to $100\%$ only in the limit of seeing infinitely many examples.

**Text Mixture Datasets.** In another set of experiments, we use text datasets such as `wikipedia` and `c4`. Our models are trained on sequences constructed from individual text samples, or combinations of 2 independent text samples coming from the source dataset. When 2 input text samples are concatenated to form a single sequence, we cut the first text excerpt at a random position sampled uniformly from the range $[l_{\text{start}}, l_{\text{finish}}]$ and concatenate the second text sequence to it. The concatenation is done after both text sequences are tokenized and the final produced sample is truncated at the maximum sequence length $l_{\text{max}}$. In most of our experiments, the total sequence length was $l_{\text{max}} = 512$, $l_{\text{start}} = 256$ and $l_{\text{finish}} = 384$.

**4.2. In-Context Learning Results**

Our first experiments were conducted with the synthetic in-context learning dataset described in Sec. 4.1 with $n_{\text{tasks}} = 4$, $n_{\text{ex}} = 4$ and $d = 3$. We trained GPT2-style Transformer models (Radford et al., 2019) following both the element-wise regularization method (Sec. 3.2) and the VAE-based approach (Sec. 3.3).

**Slow representation.** After a brief hyper-parameter tuning stage, we identified a range of parameter values, for which regularized model activations were slowly changing

and yet not collapsing to a constant value. For our element-wise regularizer, we picked $w_C$ between $0.04$ and $0.08$ and $w_D = w_C/2$. For VAE, we chose the delta-correlated noise component given by $\nu = 0.03$ or $0.1$, temporal scale given by $\sigma = 0.1$ (10% of the sequence length) and $\beta$ ranging between $0.01$ and $1.0$. Detailed model parameters can be found in Appendix C.

First, we analyzed the properties of the learned slow variables, discovering that both element-wise regularization and VAE-based approaches produced qualitatively similar results.

Fig. 4.2 shows a typical plot of the average dot-product $\langle \boldsymbol{n}_i \cdot \boldsymbol{n}_j \rangle$ of normalized slow embeddings $\boldsymbol{n}_i = \boldsymbol{y}_i/\|\boldsymbol{y}_i\|$ or of individual components $\langle n_{i,k} \cdot n_{j,k} \rangle$ emerging in most of our experiments. In that specific example we used $n_{\text{tasks}} = 4$ and $n_{\text{ex}} = 4$ so is has a global block-diagonal structure with 4 blocks corresponding to 4 independent tasks. In models trained with element-wise regularizer, the slow embedding generally evolved in the first half of each task, when the model processed the first two examples, but then stabilized and hardly changed when processing the last two examples. This is consistent with $\boldsymbol{y}$ gradually "learning" the representation of the current task $(a_i, b_i)$ while processing first 2 examples (which are generally sufficient to narrow down multiplier values).

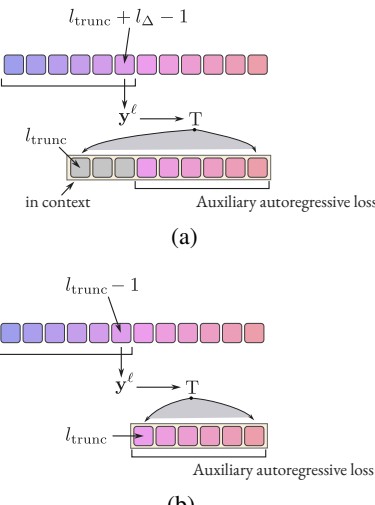

(a)

(b)

*Figure 6.* Two different training schemes for learning specialized models: **(a)** previous examples are *included* in context: the model uses $l_\Delta$ tokens both for computing the frozen replicated $\boldsymbol{y}^\ell$ and the final answer; **(b)** previous examples are *excluded* from the context when computing the final answer.

**Predicting the task from $\boldsymbol{y}$.** We also verified that $\boldsymbol{y}$ did in fact encode task multipliers $a$ and $b$ by performing linear regression. For this purpose, we computed $\boldsymbol{y}$ for each task in each sequence creating an auxiliary dataset of records $(a_i, b_i, \langle \boldsymbol{y}_i \rangle)$, where $a_i$ and $b_i$ are task parameters and $\langle \boldsymbol{y}_i \rangle$ is a value of $\boldsymbol{y}$ within this task averaged across 4 tokens (to reduce the effect of noise) in a random position in the last two examples. A typical linear fit for both of these coefficients in a model with element-wise regularization is illustrated in Fig. 4.2, where it can be seen that the predicted values of $a(\langle \boldsymbol{y} \rangle)$ and $b(\langle \boldsymbol{y} \rangle)$ with a *simple linear model* agree very well with the actual values used in the corresponding sequences. We also observed similar linear correspondence between $\boldsymbol{y}^\ell$ and the task multipliers $a$ and $b$ in VAE models, where higher $\beta$ values were typically associated with smoother $\boldsymbol{y}^\ell$. At lower values of $\beta$ and without $\boldsymbol{y}^\ell$ augmentations in the auxiliary loss, the model would typically encode some task information in rapid localized changes of $\boldsymbol{y}^\ell$ (see Appendix F).

**Model performance.** All models were compared to baseline models running on $\boldsymbol{x}$ alone without $\mathbf{T}^\nu(\cdot; \boldsymbol{y}^\ell)$ transformations. Baseline models had 6 layers, an inner dimension of 112, and 7 heads. Accuracy measurements were done on 5 or 6 independent runs, reporting the highest[2] and average test accuracy. Baseline models achieved 78.4% next-token prediction accuracy (0.2% error[3]) on numeric answers in the last 2 examples of each of 4 tasks and the average ac-

---

[2]The model is chosen based on the validation accuracy

[3]Here and later we report 3 times the standard error as our statistical error.

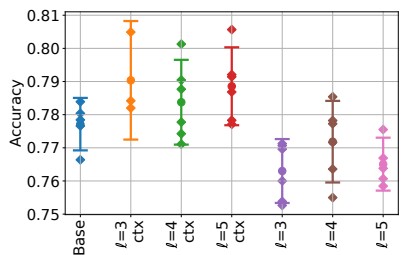

*Figure 7.* Average accuracies on the last two examples in different specialization runs for $\ell = 3, 4, 5$ (average and 3 times the standard error are also plotted): *with* previous examples in context (**ctx**) and *without* them.

curacy across all models was 77.7%. Introducing a 16- or 32-dimensional $\boldsymbol{y}^\ell$ at layer 5 and adding a regularizer with $w_C = 0.08$ and $w_D = 0.04$ increased peak accuracy to 80.4% (0.3% error) and average accuracy to 79.1%. Variational autoencoder models, without extensive hyperparameter tuning, had comparable average accuracy of 79.0% for $\beta = 0.1$. Accuracy degraded with increased regularization, but representations $\boldsymbol{y}^\ell$ became smoother (see Fig. 15(a) and discussion in Appendix F).

Even though the accuracy increase is attributed to $\boldsymbol{y}^\ell$ increasing model complexity and the total number of model parameters, our trained models can act as specialized models if we communicate the task information via $\boldsymbol{y}^\ell$. To assess model specialization via $\boldsymbol{y}^\ell$, we considered two setups, as illustrated in Fig. 6. In the first, $\boldsymbol{y}^\ell$ was computed on a sequence of task examples to generate a new specialized model with frozen $\mathbf{T}^\nu$ operators. This model, was then run on new examples of the same task with the old examples (used to generate $\mathbf{T}^\nu$) being provided in context. It reached 80.6% peak accuracy and 78.9% average accuracy for $\ell = 5$ (see Fig. 4.2), higher than the baseline accuracy.

In the second setup, $\boldsymbol{y}^\ell$ and $\mathbf{T}^\nu$ were computed similarly, but the model was applied to new samples without using the original examples in context. For $\ell = 4$, the specialized model's peak accuracy was 78.5% (0.3% error) and average accuracy was 77.3% (see Fig. 4.2). This shows that a few task examples can generate a new specialized Transformer that effectively performs the task without additional examples in context. By training our model, we effectively converted multiple task demonstrations into specialized model weights that requires no prompting or tuning.

Both setups used the same training parameters, but with $\dim \boldsymbol{y}^\ell = 64$. The auxiliary loss with $\boldsymbol{y}^\ell$ augmentation copied $\boldsymbol{y}^\ell$ across the entire sequence. In the first setup, $\tilde{\boldsymbol{y}}_k^\ell = \boldsymbol{y}_{l_\text{trunc}+l_\Delta-1}^\ell$ for $k \geq l_\text{trunc} + l_\Delta$, and in the second, $\tilde{\boldsymbol{y}}_k^\ell = \boldsymbol{y}_{l_\text{trunc}-1}^\ell$ for $k \geq l_\text{trunc}$ (see Fig. 6). This optimization allowed fixed $\mathbf{T}^\nu$ operators and accommodated previous examples in context.

### 4.3. Text Mixture Results

We began our text dataset experiments by training models using our *element-wise regularizer* and examining the properties of the learned slow features. The Transformer model we selected had 12 layers and was trained on a blend of two distinct c4 text excerpts. Similar to the synthetic dataset, we chose $w_C$ from 0.04 to 0.1, set $w_D = w_C/2$, and $\ell = 8$ (see more details in Appendix E). The emergence of clear transitions between text documents in the slow representations $\boldsymbol{y}^\ell$ of our trained models was notable, as depicted in Fig. 4.2, despite this dataset property not being explicitly utilized in our training approach. Additionally, we validated that $\boldsymbol{y}^\ell$ values serve as sensible embeddings for the current document or passage by calculating $\boldsymbol{y}$ across hundreds of wikipedia pages from 8 distinct categories (see details in Appendix E). The t-SNE (van der Maaten & Hinton, 2008) plot in Fig. 4.3 shows the clustering of pages from the same categories[4], with noticeable distinction between different categories, except for "Mathematical identities" and "Theoretical physics," which aligns with their semantic similarity. Moreover, we assessed our model on various out-of-distribution mixtures of 3 text excerpts, observing transitions of $\boldsymbol{y}$ within approximately 10 to 20 tokens from the joined text locations (see Fig. 12(b) in Appendix).

In a separate set of experiments, we trained VAE models on the c4 and wikipedia datasets. We used $\nu$ between 0.1 and 0.3, $\sigma = 0.1$, and $\beta$ from 0.1 to 10. Increasing $\beta$ made the learned slow feature $\boldsymbol{y}$ smoother. The features remained predictive of the text subject (confirmed by t-SNE plots). Very high $\beta$ values, however, resulted in overly smooth representations that couldn't reliably separate different texts (see Appendix F).

We also studied specialized models by freezing $\boldsymbol{y}^\ell$ throughout the sequence. Starting with a pre-trained model on c4 and testing on the validation set, we split 11,600 samples into two parts, typically at the end of a sentence around 400 characters (roughly 100 tokens) from the start. The model was run on the first part, then $\boldsymbol{y}^\ell$ was frozen and the model was run on the second part with $\boldsymbol{y}^\ell$ fixed, creating a specialized model. The per-token cross-entropy loss was computed and averaged over all samples. The specialized model had lower loss initially, but the non-specialized model caught up and surpassed it after roughly 200 tokens (see Fig. 4.3). This suggests that $\boldsymbol{y}^\ell$ helped the specialized model near its measured position but became less useful further down the sequence, degrading performance when forcefully fixed. Similar behavior was observed when comparing the specialized model to a separately trained baseline model, with performance leveling around 300 tokens (see Fig. 12(a) in Appendix).

---

[4]The models trained on individual documents learn even better embeddings, see Fig. 14

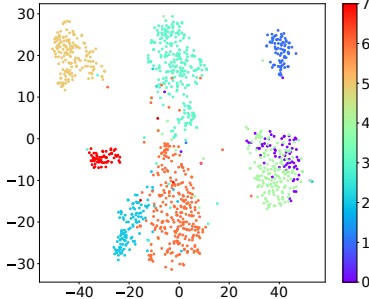

*Figure 8.* t-SNE plot for wikipedia articles from 8 different categories.

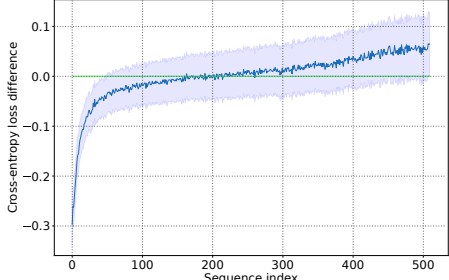

*Figure 9.* Average difference between the cross-entropy losses of the specialized (fixed $\boldsymbol{y}^\ell$) and non-specialized (dynamic $\boldsymbol{y}^\ell$) models; the specialized model is better where this difference is below zero.

Additionally, instead of clamping $\boldsymbol{y}^\ell$ to its value at the end of the first part of the document, we conducted experiments updating $\boldsymbol{y}^\ell$ as a moving average (with the values computed on the second part). We then witnessed an improved cross-entropy loss throughout the entire sequence (see Fig. 13 in Appendix) thus having a way of injecting information about the first part of the text into the second part while allowing $\boldsymbol{y}^\ell$ to adjust to shifting context. In other words, $\boldsymbol{y}^\ell$ might be interpreted as a topic embedding or *topic vector* that we can flexibly manipulate.

## 5. Discussion

Learning slow features that carry information about the global context in a sequence is important for understanding and interpreting data. Here we propose an approach for incentivizing a Transformer model to discover such slow representation within its inner activations. We then modify the model architecture to parameterize local computation by these learned slow features, showing that it is then possible to generate models that are uniquely specialized to a particular local context and no longer need to have direct access to it. While we only consider several simple examples in our experiments (a synthetic few-shot in-context learning task and a mixture of texts), we believe that this approach can prove useful for representation learning, model interpretability and generation of specialized models.

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

## A. Slowly Evolving Activations

The precise definition of what it means for activations $\boldsymbol{y}$ to change "slowly" can be defined on a case-by-case basis. For example, we could say that the sequence $\boldsymbol{y}_i$ is $K$-slow if there exist K-Lipshitz $\boldsymbol{f} : \mathbb{R} \to \mathbb{R}^d$ such that $\boldsymbol{y}_i = f(i)$. In the main text we frequently consider $\boldsymbol{y}$ to be slow if viewed across all samples, the distribution of $\boldsymbol{y}$ mimics that of a Gaussian random process with $\mathcal{K}_{i,j} = \nu\delta_{i,j} + (1 - \nu)\mathcal{K}_{i,j}^{\text{RBF}}$ with a sufficiently small $\nu$ and $\mathcal{K}_{i,j}^{\text{RBF}} = \exp(-\|i - j\|^2/2\sigma^2)$ being a Radial Basis Function (RBF) kernel with a sufficiently large $\sigma$.

Yet another family of meaningful definitions can be motivated by information theory. As a simple example, given a particular quantization scheme, we could define $\hat{n}$ as a quantized value of $\boldsymbol{n} = \boldsymbol{y}/\|\boldsymbol{y}\|$ and say that $\boldsymbol{n}$ changes slowly if the conditional entropy $\mathbb{H}(\hat{n}_i|\hat{n}_1, \ldots, \hat{n}_{i-1})$ is sufficiently small. In other words, if the new observations of $\hat{n}$ are almost pre-determined by the history, which can happen, for example, if $\boldsymbol{n}_i$ stay in individual quantization cells for a sufficiently long time.

## B. Additional Regularizers

While the element-wise regularizers described in Section 3.2 proved to be sufficiently effective in practice, we can formulate a more principled approach to regularizing $\boldsymbol{y}$. The goal of the designed regularizer is to guarantee that $\boldsymbol{y}$ seen as a random variable (considering per-sample realizations) should ideally be seen as samples from a prescribed distribution function $p(\boldsymbol{y})$. Since estimating probability distribution of a high-dimensional random process is typically complicated, we need to rely on a simpler approach. Specifically, we consider a sufficiently flexible parametric family $p_\theta$ and then regularize the values of the parameter estimators $\hat{\theta}(\boldsymbol{y})$ to be equal to their predefined values by using, for example, a regularizer

$$\mathcal{R}_P \sim \left\|\hat{\theta}(\boldsymbol{y}) - \theta_0\right\|^2. \tag{2}$$

Here we utilize a naïve $L_2$ regularization of the distribution parameters, but other choices could also be considered.

**Gaussian process example.** Instead of regularizing the derivative of $\boldsymbol{y}_s$, here we introduce a more natural constraint on $\boldsymbol{y}$ requesting that these slow activations are a stationary Gaussian process with zero mean and kernel $\mathcal{K}$ depending only on the relative position of two elements in the sequence. Different choices of $\mathcal{K}$ can control how slowly $\boldsymbol{y}_s$ is expected to change along the sequence.

Assuming that $\boldsymbol{y}$ is a multi-variate Gaussian distribution, we can estimate the mean and covariance matrix:

$$\boldsymbol{\mu}_s = \langle\boldsymbol{y}_s\rangle \quad \text{and} \quad \Sigma_{s,t} = \langle(\boldsymbol{y}_s - \boldsymbol{\mu}_s)(\boldsymbol{y}_t - \boldsymbol{\mu}_t)\rangle,$$

where the averaging is performed over the batch of samples. Remembering our Gaussian process assumption, we can then expect that $\boldsymbol{\mu}_s = 0$ and $\Sigma_{s,t,i,j} = \mathcal{K}(|s - t|)\delta_{i,j}$, which we can enforce by utilizing the regularizer (2):

$$\mathcal{R}_P \sim \frac{1}{N}\sum_s \|\boldsymbol{\mu}_s\|^2 + \frac{1}{N^2}\sum_{s,t,i,j}\left(\langle\Delta y_{s,i}\Delta y_{t,j}\rangle_\alpha - \mathcal{K}_{|s-t|}\delta_{i,j}\right)^2,$$

where $N$ is the total number of elements in each sequence and $\Delta\boldsymbol{y}_s := \boldsymbol{y}_s - \boldsymbol{\mu}_s$. Here $\langle\cdot\rangle$ denotes averaging over individual samples in the batch. Notice that in practice, we can reduce the cost of the proposed computation by sampling only a small set of all possible sequence elements $(s, t)$ or embedding dimensions $(i, j)$.

Notice that we can also use a simplified form of this regularizer, where we remove constraints on cross-token correlations:

$$\mathcal{R}'_D \sim \sum_s \left[\|\langle\boldsymbol{y}_s\rangle\|^2 + \sum_{i,j}\left(\langle\Delta y_{s,i}\Delta y_{s,j}\rangle - \delta_{i,j}\right)^2\right],$$

where $\Delta\boldsymbol{y} := \boldsymbol{y} - \langle\boldsymbol{y}\rangle$ and $\langle\cdot\rangle$ denotes averaging over individual elements in a batch. Compared to the orthogonal projection loss used in Section 3.1, here we instead compute and regularize sample statistics.

## C. Model Details and Parameters

In all of our experiments, we used GPT-2 style Transformer models with GELU nonlinearities.

Each MLP layer separated $x$ and $y$ transformations, effectively using two MLPs for processing $x$ and $y$ correspondingly:

$$x^{\nu+1} = \mathbf{W}_2^{x}\sigma(\mathbf{W}_1^{x}x^{\nu}),$$
$$y^{\nu+1} = \mathbf{W}_2^{y}\sigma(\mathbf{W}_1^{y}[x^{\nu}, y^{\nu}]),$$

where $[\cdot, \cdot]$ denotes vector concatenation, $\mathbf{W}_*^*$ are linear operators with corresponding matrices $\hat{\mathbf{W}}_1^{x} \in \mathbb{R}^{i_x \times d_x}$, $\hat{\mathbf{W}}_2^{x} \in \mathbb{R}^{d_x \times i_x}$, $\hat{\mathbf{W}}_1^{y} \in \mathbb{R}^{i_y \times (d_x + d_y)}$, $\hat{\mathbf{W}}_2^{y} \in \mathbb{R}^{d_y \times i_y}$. The inner dimensions were typically chose to be $i_x := 4d_x = 4\dim x$ and $i_y := 4d_y = 4\dim y$.

Similarly, each self-attention layer had separate $H_x$ heads acting on $x$ alone and producing the final output that was completely $y$-independent. Total of $H_y$ $(d_y/H_y)$-dimensional heads were reserved for self-attention on $y$ with key/query/value vectors generated from the complete state $(x, y)$ thus allowing $y$ to absorb information from $x$:

$$k^{x} = \mathbf{K}^{x}x, \qquad\qquad q^{x} = \mathbf{Q}^{x}x, \qquad\qquad v^{x} = \mathbf{V}^{x}x,$$
$$k^{y} = \mathbf{K}^{y}[x, y], \qquad\qquad q^{y} = \mathbf{Q}^{y}[x, y], \qquad\qquad v^{y} = \mathbf{V}^{y}[x, y],$$

where we omitted the computation stage index $\nu$ and the head index $h$ for brevity.

Each Transformer block contained self-attention layer followed by the MLP layer, as described above, with inner normalizaton operations applied separately to $x$ and $y$.

Before layer $\ell$, the computation on $x$ was completely independent of $y$, but at and after layer $\ell$, we applied an additional transformation $\mathbf{T}^{l}(x^{l}; y^{\ell})$ on $x^{l}$ before each self-attention and each MLP operation.

### C.1. Model Parameters

We trained our models using ADAM optimizer with the learning rate typically set to $2.5 \cdot 10^{-4}$ or $5 \cdot 10^{-4}$ for the total of 400,000 steps with cosine learning rate decay (warmup of 10,000 steps) and batch size of 128. We used Google TPU v5e 4x4 as our training hardware platform, which took us to spend about 10 hours training the model. We did not use dropout in most of our experiments, which allowed us to reach higher accuracies in the in-context learning setup, but resulted in a degraded model stability: the final model accuracy for different initial seeds could differ by as much as 2%. High values of weight decay were also observed to hurt the model performance and we set it to $10^{-8}$ in most of our experiments.

**In-Context Learning.** In most of our experiments with the synthetic dataset, we used a 6-layer model with 7 to 11 self-attention heads. The baseline model had $h_x = 7$ heads with the embedding size of $d_x = 112$. The model with $d_y$-dimensional $y^{\ell}$ used $7 + h_y$ heads, where $h_y = d_y/16$, making the total embedding size equal to $d_x + d_y = 112 + 16h_y$. In our experiments with specialized models, we chose $d_y = 64$ (and hence $h_y = 4$) and the rank of $\delta\hat{\mathbf{W}}^{\nu}$ was 4 and $M = 16$ (total number of $\hat{L}$ and $\hat{R}$ matrices). We chose $w_C = 0.08$ and $w_D = 0.04$. When using augmentations with auxiliary losses, we typically chose $l_{\Delta} = 30$ and computed the auxiliary loss on the entire sequence from $l_{\text{trunc}} + l_{\Delta}$ to $n$ with $n$ being the total sequence length and $l_{\text{trunc}}$ being sampled uniformly from $[0, (3/4)n - l_{\Delta}]$. We also experimented with smaller-sized contexts in auxiliary loss where the cross-attention was only computed on $[l_{\text{trunc}} + l_{\Delta}, l_{\text{trunc}} + l_{\Delta} + l_{\omega}]$ with $l_{\omega}$ being equal to $n/4$. Choosing this smaller sequence length increase loss variance, but appeared to improve the average model performance by 0.2%.

**Ablation studies.** We conducted additional ablation studies varying three parameters:

1. rank $r$ of the generated matrix (Fig. 10(a)),

2. value of $w_C$ with $w_D = w_C/2$ (Fig. 10(b)),

3. value of $w_D$ for a fixed $w_C = 0.08$ (Fig. 10(c)).

All experiments measured the performance of specialized models with $\dim y^{\ell} = 64$ with examples used to generate $y^{\ell}$ presented *in context* (*first setup* in Sec. 4.2). While it is clear that confident statements require significantly more experiments for statistically significant results, we may draw some preliminary conclusions. Firstly, models with generated rank-2 and rank-4 matrices appear to outperform models with rank-1 matrices. Increasing derivative regularization strength $w_C$ appears to hurt performance above $w_C = 0.04$. And increasing the orthogonal projection loss weight $w_D$ appears to not hurt model performance and possibly even improves it.

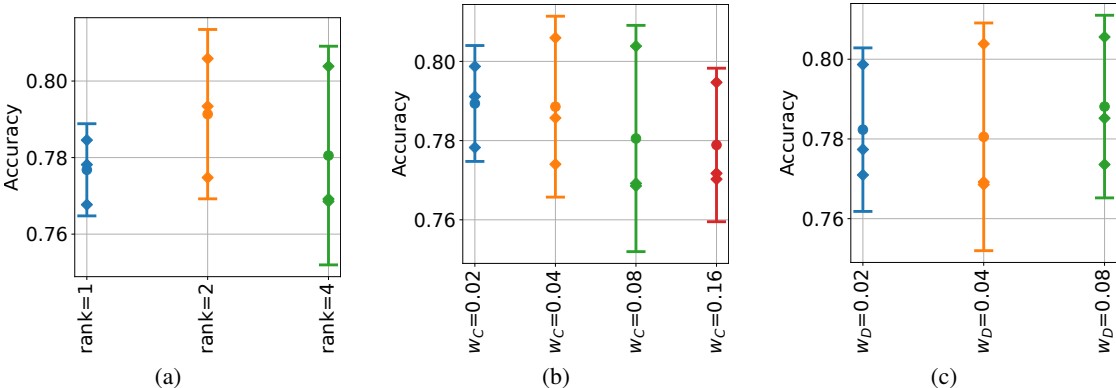

*Figure 10.* Ablation study results: **(a)** varying the rank of the generated matrices with $w_C = 2w_D = 0.08$; **(b)** varying $w_C$ with $w_D = w_C/2$; **(c)** varying $w_D$ for a fixed $w_C = 0.08$.

**Text Mixture.** In our text experiments, we chose $w_C = 2w_D = 0.08$ and our models contained 12 layers with $\ell = 8$. The total number of tokens was equal to 8000 byte pair encoding subwords (Sennrich et al., 2015) and the total sequence size was 512.

**VAE Parameters.** Our VAE model was typically trained with $\nu = 0.03$ in the in-context learning setup and $0.1$ in text datasets. The characteristic auto-correlation size was chosen as $\sigma = 0.1$ (10% of the sequence length) and $\beta$ varied from $0.01$ to $10.0$.

## D. In-Context Learning: Additional Details

### D.1. Dataset Examples

In our experiments, we typically chose $n_{\text{tasks}} = 4$ with $n_{\text{ex}} = 4$, or $n_{\text{tasks}} = 1$ with $n_{\text{ex}} = 8$. An example of a generated ASCII sequence before tokenization is:

```
154*709=+07058|648*011=+05920|526*187=+06230|893*495=+11997|#
122*395=-00273|827*301=+00526|216*082=+00134|399*879=-00480|#
913*075=+01063|748*228=+01204|508*205=+00918|186*523=+01232|#
349*703=+04547|343*849=+04785|868*591=+08994|124*356=+01828|#
```

All these lines concatenated together form a single sample. Here we put different tasks on different lines for clarity.

### D.2. Additional Experimental Results

In addition to our experiments with $n_{\text{tasks}} = 4$ and $n_{\text{ex}} = 4$, we also conducted experiments using a single-task dataset ($n_{\text{tasks}} = 1$) with $n_{\text{ex}} = 8$ examples. The plot of the dot-product $\boldsymbol{n}_i \cdot \boldsymbol{n}_j$ (see Fig. 11(b)) can again be seen to reflect a gradual convergence of $\boldsymbol{y}$ as more and more examples are being processed (see Fig. 11(a)). Here we used a larger baseline model with 8 layers instead of 6, which reached the top accuracy of 82.7%. We then verified that multiple models trained with element-wise regularization, 32-dimensional $\boldsymbol{y}$, $\ell$ being 4 or 5, softmax-based rank-4 matrix generator, our auxiliary loss and augmentation methods were able to achieve accuracies in the range 82.6% to 82.8% while using a *frozen* value of $\boldsymbol{y}$ obtained using several samples from the same task.

## E. Text Mixture Dataset: Additional Details

**8 different categories** "Mathematical identities" (0), "Real-time operating systems" (1), "Songs about nights" (2), "American abstract artists" (3), "Theoretical physics" (4), "State parks of Washington (state)" (5), "Film genres" (6) and "Three-ingredient cocktails" (7).

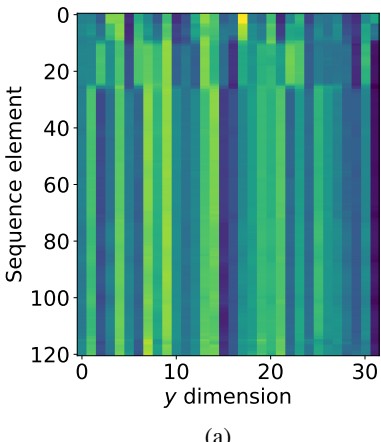

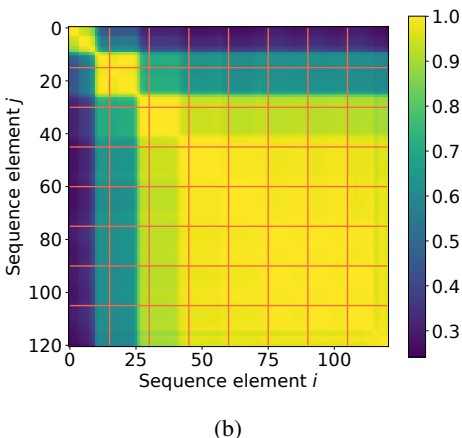

(a)                           (b)

*Figure 11.* **(a)** A typical dependence on $\boldsymbol{y}^{\ell}$ on the sequence index for a synthetic in-context learning task with 8 examples; **(b)** dot product $\boldsymbol{n}_i \cdot \boldsymbol{n}_j$ for normalized $\boldsymbol{y}^{\ell}$ embeddings at two different locations for this synthetic dataset.

**Sample composed of 3 text excerpts.** Phrase composed of 3 different texts used in our experiments for verifying transitions of $\boldsymbol{y}$ (see Fig. 12(b)):

> The horned sungem (Heliactin bilophus) is a species of hummingbird native to much of central Brazil and parts of Bolivia and Suriname. It prefers open habitats such as savanna and grassland and readily occupies human-created habitats such as gardens. It recently expanded its range into southern Amazonas and Espirito Santo, probably as a result of deforestation; few other hummingbird species have recently expanded their range. The horned sungem is a small hummingbird with a long tail and a comparatively short, black bill. The sexes differ markedly in appearance, with males sporting two feather tufts ('horns') above the eyes that are shiny red, golden, and green. Linux was originally developed for personal computers based on the Intel x86 architecture, but has since been ported to more platforms than any other operating system. Because of the dominance of Linux-based Android on smartphones, Linux, including Android, has the largest installed base of all general-purpose operating systems as of May 2022. Linux is, as of March 2024, used by around 4 percent of desktop computers, the Chromebook, which runs the Linux kernel-based ChromeOS, dominates the US K–12 education market and represents nearly 20 percent of sub-$300 notebook sales in the US. Horse races vary widely in format, and many countries have developed their own particular traditions around the sport. Variations include restricting races to particular breeds, running over obstacles, running over different distances, running on different track surfaces, and running in different gaits. In some races, horses are assigned different weights to carry to reflect differences in ability, a process known as handicapping. Horse racing has a long and distinguished history and has been practiced in civilizations across the world since ancient times. Archaeological records indicate that horse racing occurred in Ancient Greece, Ancient Rome, Babylon, Syria, Arabia, and Egypt.

**Token probabilities.** We conducted additional experiments with specialized language models obtained by freezing $\boldsymbol{y}^{\ell}$ value to a constant throughout the sequence. Specifically, we verified that replacing $\boldsymbol{y}^{\ell}$ for one sequence with $\boldsymbol{y}^{\ell}$ values from a different sequence has an expected impact on output token likelihoods. For example, by using $\boldsymbol{y}^{\ell}$ from a "Theoretical Physics" page on a text from "American abstract artists" category, we observe that among top 500 tokens, the logits of "engine", "theory", "mechanics", "science", "condit", "chem", "physics" and other similar tokens, increased the most on average.

## F. VAE Results

The effect of varying $\beta$ in our VAE experiments with the in-context few-shot learning dataset are shown in Figure 15. We trained multiple models with different values of $\beta$ and observed that the model with $\beta = 0.01$ and hence virtually

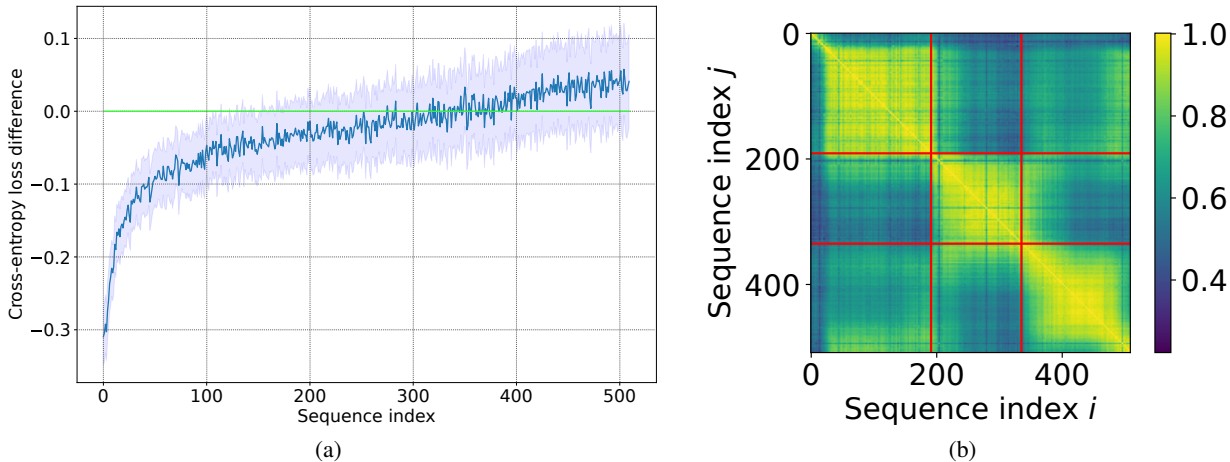

(a)         (b)

*Figure 12.* **(a)** average difference (on the second part of the document) between cross-entropies of a specialized model with $\boldsymbol{y}^\ell$ pre-computed on the first part and a baseline language model; **(b)** dot-product plot $\boldsymbol{n}_i \cdot \boldsymbol{n}_j$ for a combination of 3 different text excerpts described in Appendix E (with boundaries shown).

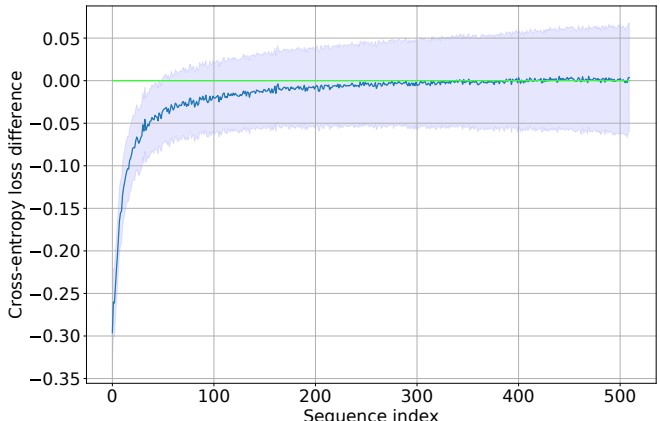

*Figure 13.* Average difference between the cross-entropy losses of the "informed" and "uninformed" (non-specialized) models. The informed model is generally better across the entire sequence (the difference is below zero). The informed model used dynamic value of $\boldsymbol{y}^\ell$ initialized with $(\boldsymbol{y}^\ell)^{\text{init}}$ computed at the end of the first part and then maintained with a moving average with the rate $\gamma = 1/300$. In other words, we used $(\boldsymbol{y}^\ell)_i^{\text{used}} = (1-\gamma)(\boldsymbol{y}^\ell)_{i-1}^{\text{used}} + \gamma(\boldsymbol{y}^\ell)_i^{\text{computed}}$ with $(\boldsymbol{y}^\ell)_0^{\text{used}} = (\boldsymbol{y}^\ell)^{\text{init}}$. Maintaining this moving average allowed us to utilize information about the topic of the first part of the text without freezing $\boldsymbol{y}^\ell$ throughout the entire sequence. The uninformed model maintained a dynamic computed $\boldsymbol{y}^\ell$ without any direct or indirect access to the first part of the text.

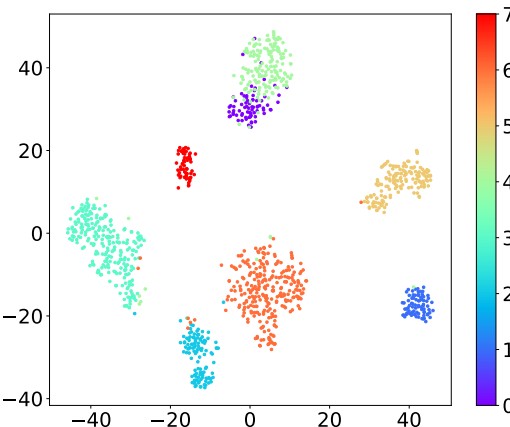

*Figure 14.* t-SNE plot for pages from 8 `wikipedia` categories using a model trained on individual `c4` articles instead of pairs of randomly joined samples. This plot shows a much better separation between different categories, which is probably due to this test distribution being closer to the training set distribution (where each sample was generally touching a single topic).

non-existent KL divergence term exhibited strong periodicity (on task boundaries), but as we increased $\beta$, model activations $\boldsymbol{y}^\ell$ became smoother (see Fig. 15(a)). Also, while for smaller $\beta$, the model tended to encode some task information in rapidly changing activation components, this behavior almost vanished at higher values of $\beta$ and model activations became a good predictor of the task multipliers $a$ and $b$. The effect of $\beta$ on model accuracy was also unsurprising in that strong regularization with higher values of $\beta$ appeared to hurt model performance (see Fig. 15(b)) suggesting that there might be a minor conflict between learning maximally useful representations $\boldsymbol{y}^\ell$ and these representations adhering perfectly to our desired prior.

Additional VAE results with `c4` dataset and varying values of $\beta$ are presented in Fig. 16, 17 and 18. First we show the dot-product $\boldsymbol{n}_i \cdot \boldsymbol{n}_j$ on a mixture of 3 distinct texts described in Appendix E for different values of $\beta$ (Fig. 16). We then illustrate t-SNE plots of learned features on 8 distinct `wikipedia` categories (Fig. 17). Finally, in Fig. 18, we show traces of $\boldsymbol{y}^\ell$ activations on a mixture of 3 texts. It can be seen that increasing $\beta$ makes learned slow activations much smoother.

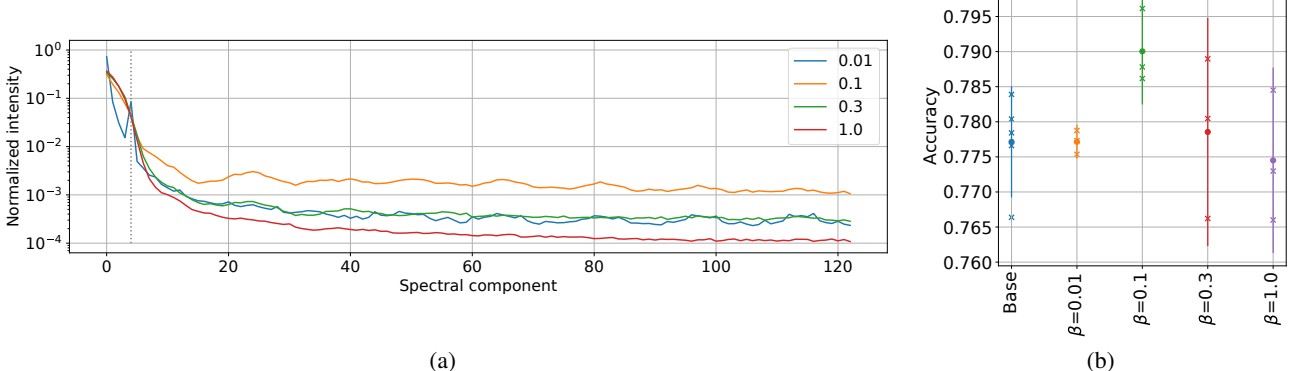

(a)                                           (b)

*Figure 15.* **(a)** Normalized averaged intensity $\langle |f_k|^2 \rangle$ of discrete Fourier transform spectra $f_k$ of all $\boldsymbol{y}^\ell$ components for VAEs with $\beta$ equal to 0.01, 0.1, 0.3 and 1.0. The averaging is performed over all components of $\boldsymbol{y}^\ell$ and over 256 samples. The averaged intensity is then normalized to 1 for each experiment for comparison. The model with $\beta = 0.01$ can be seen to have a peak around the 4$^{\text{th}}$ harmonic. As $\beta$ increases, the spectrum smooths and higher harmonics disappear; **(b)** Model accuracies measured for the last 2 examples in VAE models with different $\beta$ values (showing individual accuracies, means and $3\sigma$).

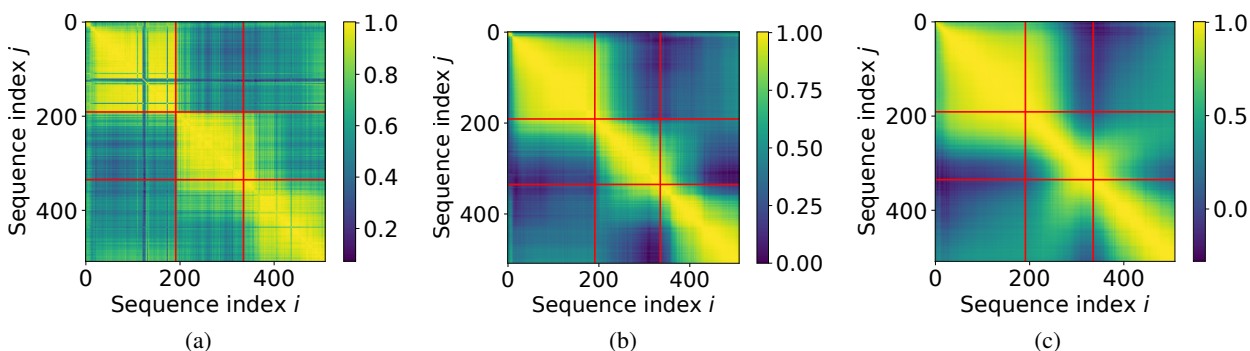

(a)                                  (b)                                  (c)

*Figure 16.* Dot product $\boldsymbol{n}_i \cdot \boldsymbol{n}_j$ plot computed for 3 different VAE models trained on `c4` and evaluated on a mixture of 3 distinct texts (see Appendix E): (a) $\beta = 1$, (b) $\beta = 3$, (c) $\beta = 10$.

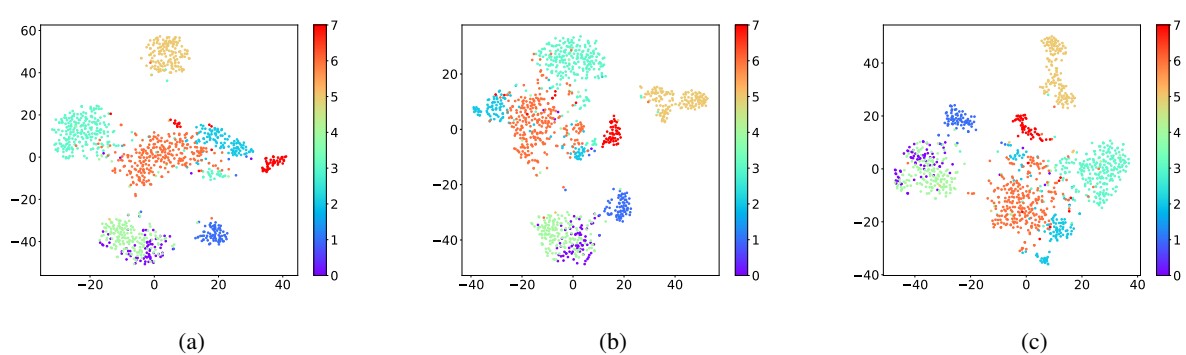

(a)                                  (b)                                  (c)

*Figure 17.* t-SNE plots for 3 different VAE models trained on `c4` and evaluated on `wikipedia` pages from 8 distinct categories: (a) $\beta = 1$, (b) $\beta = 3$, (c) $\beta = 10$.

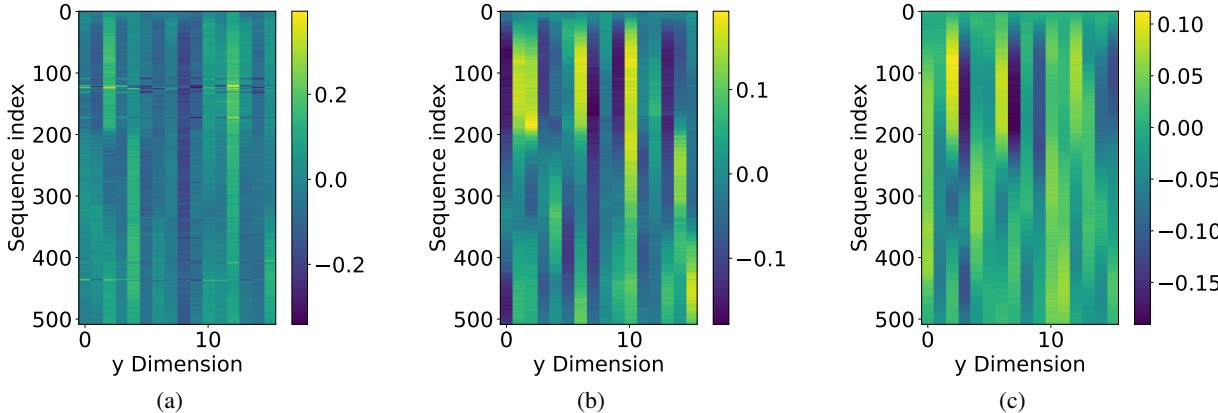

*Figure 18.* Slow activation $\boldsymbol{y}$ evolution along the sequence for 3 different VAE models trained on `c4` and evaluated on a mixture of 3 distinct texts (see Appendix E): (a) $\beta = 1$, (b) $\beta = 3$, (c) $\beta = 10$.

