# OpenReview forum: "Learning Fast and Slow: Representations for In-Context Weight Modulation"
_ICML.cc/2024/Workshop/ICL — ICML 2024 Workshop ICL Poster_

### Official Review · Reviewer_uh9s · 2024-06-05
**Decoupling high-level context from instance-level information in transformer representations**

**Rating:** 2
**Fit:** 3
**Confidence:** 2

**Workshop Review:**

**Summary:**
* This paper proposes a method to decouple "slow" and "fast" learned representations in transformer embeddings and finds that slow representations are able to capture high-level context while fast representations are more helpful for specific instances.

**Clarity/Correctness:**
* The motivation for disentangling the two hypothesized representations is well presented. The explanation of how to use regularization techniques to encourage the learning of a slow representation by the model was also thoroughly discussed.
* It was unclear to me whether the results in Figure 7 are significant. For the synthetic task introduced, the model has base performance of ~78%, while incorporating $y^{\ell}$ seems to improve performance to ~80%. It doesn't seem like much of an increase, but maybe I misunderstood the units/what is being measured here.
* It was a bit unclear when results were reported for VAE vs. the regularization approach. I think if the results could be presented together or contrasted a bit more that might be helpful to disambiguate the two methods.
* The slow representations seem to cluster well by context, which supports the idea that the model can also use it to differentiate between contexts
* It is unclear whether longer context lengths will make results presented in figure 9 worse. The CE loss difference seems to be increasing instead of converging around 0.

**Novelty/Interest:**
* Understanding network representations during ICL is a topic the community is interested in. I think the evidence presented showing that high-level contextual information is able to be decoupled from specific instance information in LM representations and can transfer to other settings to change topics is an interesting finding and may be related to recent work showing LM hidden states of ICL prompts compress & contain generalizing high-level task information [1,2]

[1] Hendel, Roee, Mor Geva, and Amir Globerson. "In-Context Learning Creates Task Vectors." Findings of the Association for Computational Linguistics: EMNLP 2023. 2023.

[2] Todd, Eric, et al. "Function Vectors in Large Language Models." International Conference on Learning Representations (ICLR 2024), 2024.

**Minor Notes:**
* The figure references in the paper are not consistent with the figure numbering. There are text references to "Figure 4.2", or "Fig. 4.3" for example, but I think it might be referring to Figure 5 or 6?

**Reason For Not Giving Higher Score:**

* The evaluation is limited to small models on a few datasets (synthetic math & mixed natural text)
* Some things were unclear in the presentation of the method/evaluation

**Reason For Not Giving Lower Score:**

* The topic is of interest to the community
* The paper has interesting results, which seem to show their decoupling objective allows slow-information to be represented adequately and separate from the instance-level representation

---

### Official Review · Reviewer_KNUz · 2024-06-13

**Rating:** 2
**Fit:** 3
**Confidence:** 2

**Workshop Review:**

This paper introduces a method to disentangle slow-evolving global context representations from fast-changing local representations in activations. The authors propose regularization techniques and a VAE-based approach to encourage the emergence of slow features, and demonstrate their effectiveness. The slow representations are then used to modulate the weights of the model, which gives specialized models for specific contexts

### Clarity/Correctness:
The experiments on the synthetic in-context learning dataset and text mixture datasets provide evidence for the emergence of slow features that capture global context information. The results show that the slow representations can predict task parameters in the synthetic dataset and cluster by topic in the text mixture datasets.
The results are presented clearly, with informative visualizations and tables. The authors provide insights into the learned slow representations, demonstrating their ability to capture global context information, Fig 5, 7.
One area that could be improved is the discussion of the limitations and potential drawbacks of the proposed method. For example, the authors could elaborate on the trade-offs between the element-wise regularization and VAE-based approaches, and discuss the scalability of their method to larger models and datasets.

### Novelty/Interest:
The problem of disentangling global context and local information in representations is a novel in ICL community
The proposed method of using learned slow features to modulate model weights and generate specialized models is a creative and potentially impactful contribution.

**Reason For Not Giving Higher Score:**

The experimental evaluation could be more comprehensive. It would be better to see how the approach performs on a wider range of tasks and datasets

**Reason For Not Giving Lower Score:**

The paper provides an insight on fast-slow representation in ICL, which is naturally suitable for ICL task and this problem is important

---

### Meta-Review · Area_Chair_RFst · 2024-06-17

**Recommendation:** 2

**Metareview:**

Reviewers agree that this is a nice paper on structuring in-context recommendations with a generative model to capture information evolving at separate timescales, though that the evaluation is limited in its current form.

---

### Decision · Program_Chairs · 2024-06-17

Accept (Poster)